# Deep Discriminative to Kernel Generative Networks for Calibrated Inference

## Abstract

The fight between discriminative versus generative goes deep, in both the study of artificial and natural intelligence. In our view, both camps have complementary values. So, we sought to synergistically combine them. Here, we propose a methodology to convert deep discriminative networks to kernel generative networks. We leveraged the fact that deep models, including both random forests and deep networks, learn internal representations which are unions of polytopes with affine activation functions to conceptualize them both as generalized partitioning rules. We replace the affine function in each polytope populated by the training data with Gaussian kernel that results in a generative model. Theoretically, we derive the conditions under which our generative models are a consistent estimator of the corresponding class conditional density. Moreover, our proposed models obtain well calibrated posteriors for in-distribution, and extrapolate beyond the training data to handle out-of-distribution inputs reasonably. We believe this approach may be an important step in unifying the thinking and the approaches across the discriminative and the generative divide.

## 1   Introduction

Machine learning methods, specially deep neural networks and random forests have shown excellent performance in many real-world tasks, including drug discovery, autonomous driving and clinical surgery. However, calibrating confidence over the whole feature space for these models remains a key challenge in the field. Although these learning algorithms can achieve near optimal performance at inferring on the samples lying in the high density regions of the training data [1–3], they yield highly confident predictions for the samples lying far away from the training data [4]. Calibrated confidence within the training or in-distribution (ID) region as well as in the out-of-distribution (OOD) region is crucial for safety critical applications like autonomous driving and computer-assisted surgery, where any aberrant reading should be detected and taken care of immediately [4, 5]. A well-calibrated model capable of quantifying the uncertainty associated with inference for any points from the training distribution as well as detecting OOD data can be a life-saver in these cases.

The approaches to calibrate OOD confidence for learning algorithms described in the literature can be roughly divided into two groups: discriminative and generative. Discriminative approaches try to scale the posteriors based on OOD detection or modify the learning loss function. Intuitively, the easiest solution for OOD confidence calibration is to learn a function that gives higher scores for in-distribution samples and lower scores for OOD samples, and thereby re-scale the posterior or confidence score from the original model accordingly [6]. There are a number of approaches in the literature which try to either modify the loss function [7–9] or adversarially train the network to be less confident on OOD samples [10, 4]. However, one can adversarially manipulate an OOD sample where the model is less confident to find another OOD sample where the model is overconfident [11, 4, 12]. Recently, as shown by Hein et al. [4], the ReLU networks produce arbitrarily high

confidence as the inference point moves far away from the training data. Therefore, calibrating ReLU networks for the whole OOD region is not possible without fundamentally changing the network architecture. As a result, all of the aforementioned algorithms are unable to provide any guarantee about the performance of the network through out the whole feature space. On the other end of the spectrum, the generative group tries to learn generative models for both the in-distribution as well as the out-of-distribution samples. The general idea for the generative group is to get likelihoods for a particular sample out of the generative models for both ID and OOD to do likelihood ratio test [13] or control the likelihood for training distribution far away from the training data to detect OOD samples by thresholding. However, it is not obvious how to control likelihoods far away from the training data for powerful generative models like variational autoencoders (VAEs) [14] and generative adversarial networks (GAN) [15]. Moreover, Nalisnick et al. [16] and Hendrycks et al. [10] showed VAEs and GANs can also yield overconfident likelihoods far away from the training data.

The algorithms described so far are concerned with OOD confidence calibration for deep-nets only. However, in this paper, we show that other approaches which partition the feature space, for example random forest, can also suffer from poor confidence calibration both in the ID and the OOD regions. Moreover, the algorithms described above are concerned about the confidence of the algorithms in the OOD region only and they do not address the confidence calibration within the training distribution at all. This issue is addressed separately in a different group of literature [17–19]. In this paper, we consider both calibration problem jointly and propose an approach that achieves good calibration throughout the whole feature space.

In this paper, we conceptualize both random forest and ReLU networks as generalized partitioning rules with an affine activation over each polytope. We consider replacing the affine functions learned over the polytopes with Gaussian kernels. We propose two novel kernel density estimation techniques named *Kernel Generative Forest* (KGF) and *Kernel Generative Network* (KGN). We theoretically show that they asymptotically converge to the true training distribution under certain conditions. At the same time, the estimated likelihood from the kernel generative models decreases for samples far away from the training samples. By adding a suitable bias to the kernel density estimate, we can achieve calibrated posterior over the classes in the OOD region. It completely excludes the need for providing OOD training examples to the model. We conduct several simulation and real data studies that show both KGF and KGN are robust against OOD samples while they maintain good performance in the in-distribution region.

## 2   Related Works and Our Contributions

There are a number of approaches in the literature which attempt to learn a generative model and control the likelihoods far away from the training data. For example, Ren et al. [13] employed likelihood ratio test for detecting OOD samples. Wan et al. [8] modify the training loss so that the downstream projected features follow a Gaussian distribution. However, there is no guarantee of performance for OOD detection for the above methods. To the best of our knowledge, only Meinke et al. [5] has proposed an approach to guarantee asymptotic performance for OOD detection. They model the training and the OOD distribution using Gaussian mixture models which enable them to control the class conditional posteriors far away. Compared to the aforementioned methods, our approach differs in several ways:

- We address the confidence calibration problems for both ReLU-nets and random forests from a common ground.

- We address in-distribution (ID) and out-of-distribution (OOD) calibration problem as a continuum rather than two separate problems.

- We provide guarantees for asymptotic convergence of our proposed approach under certain conditions for both ID and OOD regions.

- We propose an unsupervised OOD calibration approach, i.e., we do not need to train exhaustively on different OOD samples.

## 3 Methods

### 3.1 Setting

Consider a supervised learning problem with independent and identically distributed training samples $\{(\mathbf{x}_i, y_i)\}_{i=1}^n$ such that $(X, Y) \sim P_{X,Y}$, where $X \sim P_X$ is a $\mathcal{X} \subseteq \mathbb{R}^d$ valued input and $Y \sim P_Y$ is a $\mathcal{Y} = \{1, \cdots, K\}$ valued class label. We define in-distribution region as the high density region of $P_{X,Y}$ and denote it by $\mathcal{S}$. Here the goal is to learn a confidence score, $\mathbf{g} : \mathbb{R}^d \to [0, 1]^K$, $\mathbf{g}(\mathbf{x}) = [g_1(\mathbf{x}), g_2(\mathbf{x}), \ldots, g_K(\mathbf{x})]$ such that,

$$g_y(\mathbf{x}) = \begin{cases} P_{Y|X}(y|\mathbf{x}), & \text{if } \mathbf{x} \in \mathcal{S} \\ P_Y(y), & \text{if } \mathbf{x} \notin \mathcal{S} \end{cases}, \quad \forall y \in \mathcal{Y} \tag{1}$$

where $P_{Y|X}(y|\mathbf{x})$ is the posterior probability for class $y$ given by the Bayes formula:

$$P_{Y|X}(y|\mathbf{x}) = \frac{P_{X|Y}(\mathbf{x}|y)P_Y(y)}{\sum_{k=1}^K P_{X|Y}(\mathbf{x}|k)P_Y(k)}, \quad \forall y \in \mathcal{Y}. \tag{2}$$

Here $P_{X|Y}(\mathbf{x}|y)$ is the class conditional density for the training data which we will refer as $f_y(\mathbf{x})$ hereafter for brevity.

### 3.2 Background and Main Idea

Deep discriminative networks partition the feature space $\mathbb{R}^d$ into a union of $p$ affine polytopes $Q_r$ such that $\bigcup_{r=1}^p Q_r = \mathbb{R}^d$, and learn an affine function over each polytope [4, 20]. Mathematically, the class-conditional density for the label $y$ estimated by these deep discriminative models at a particular point $\mathbf{x}$ can be expressed as:

$$\hat{f}_y(\mathbf{x}) = \sum_{r=1}^p (\mathbf{a}_r^\top \mathbf{x} + b_r) \mathbb{1}(\mathbf{x} \in Q_r). \tag{3}$$

For example, in the case of a decision tree, $\mathbf{a}_r = \mathbf{0}$, i.e., decision tree assumes uniform distribution for the class-conditional densities over the leaf nodes. Among these polytopes, the ones that lie on the boundary of the training data extend to the whole feature space and hence encompass all the OOD samples. Since the posterior probability for a class is determined by the affine activation over each of these polytopes, the algorithms tend to be overconfident when making predictions on the OOD inputs. Moreover, there exist some polytopes that are not populated with training data. These unpopulated polytopes serve to interpolate between the training sample points. If we replace the affine activation function of the populated polytopes with Gaussian kernel $\mathcal{G}$ learned using maximum likelihood approach on the training points within the corresponding polytope and prune the unpopulated ones, the tail of the kernel will help interpolate between the training sample points while assigning lower likelihood to the low density or unpopulated polytope regions of the feature space. This may result in better confidence calibration for the proposed modified approach.

### 3.3 Proposed Model

Consider the collection of polytope indices $\mathcal{P}$ which contains the indices of total $\tilde{p}$ polytopes populated by the training data. We consider replacing the affine function over the populated polytopes with a Gaussian kernel $\mathcal{G}(\cdot; \hat{\mu}_r, \hat{\Sigma}_r)$. For a particular inference point $\mathbf{x}$, we consider the Gaussian kernel with the minimum distance from the center of the kernel to the corresponding point:

$$r_\mathbf{x}^* = \underset{r}{\operatorname{argmin}} \|\mu_r - \mathbf{x}\|, \tag{4}$$

where $\|\cdot\|$ denotes a suitable distance measure. We use Euclidean distance metric while conducting the simulation and the benchmark datasets experiments in this paper for simplicity. In short, we modify Equation 3 from the parent ReLU-net or random forest to estimate the class-conditional density as:

$$\tilde{f}_y(\mathbf{x}) = \frac{1}{n_y} \sum_{r \in \mathcal{P}} n_{ry} \mathcal{G}(\mathbf{x}; \mu_r, \Sigma_r) \mathbb{1}(r = r_\mathbf{x}^*), \tag{5}$$

where $n_y$ is the total number of samples with label $y$ and $n_{ry}$ is the number of samples from class $y$ that end up in polytope $Q_r$. We add a bias to the class conditional density $\tilde{f}_y$:

$$\hat{f}_y(\mathbf{x}) = \tilde{f}_y(\mathbf{x}) + \frac{b}{\log(n)}. \tag{6}$$

Note that in Equation 6, $\frac{b}{\log(n)} \to 0$ as the total training points, $n \to \infty$. The class posterior probability (confidence) $\hat{g}_y(\mathbf{x})$ of class $y$ for a test point $\mathbf{x}$ is estimated using the Bayes rule as follows:

$$\hat{g}_y(\mathbf{x}) = \frac{\hat{f}_y(\mathbf{x})\hat{P}_Y(y)}{\sum_{k=1}^{K} \hat{f}_k(\mathbf{x})\hat{P}_Y(k)}, \tag{7}$$

where $\hat{P}_Y(y)$ is the empirical prior probability of class $y$ estimated from the training data. We estimate the class for a particular inference point $\mathbf{x}$ as:

$$\hat{y} = \operatorname*{argmax}_{y \in \mathcal{Y}} \hat{g}_y(\mathbf{x}). \tag{8}$$

### 3.4 Desiderata

We desire our proposed model to estimate confidence score $\hat{g}_y$ to satisfy the following two desiderata:

1. **Asymptotic Performance**: We want point-wise convergence for our estimated confidence as $n \to \infty$, i.e.,

$$\max_{y \in \mathcal{Y}} \sup_{\mathbf{x} \in \mathbb{R}^d} |g_y(\mathbf{x}) - \hat{g}_y(\mathbf{x})| \to 0.$$

2. **Finite Sample Performance**: We want better posterior calibration for $\hat{g}_y(\mathbf{x})$ both in ID and OOD region compared to that of its parent model.

We theoretically derive the conditions under which we achieve Desiderata 1 in Section 4. However, we run extensive experiments on various simulation and benchmark datasets in Section 6 to empirically verify that our proposed approach achieves Desiderata 2.

## 4 Theoretical Results

**Theorem 1** (Asymptotic Convergence to the True Distribution). *Consider a partition rule that partitions $\mathbb{R}^d$ into hypercubes of the same size $h_n > 0$. Formally, let $\mathcal{P}_n = \{Q_1, Q_2, \cdots\}$ be a partition of $\mathbb{R}^d$, that is, it partitions $\mathbb{R}^d$ into sets of the type $\Pi_{i=1}^{d}[\psi_i h_n, (\psi_i + 1)h_n)$, where $\psi_i$'s are integers. Let $n$ be the total number of samples and $n_r$ be the number of data points within polytope $Q_r$. Consider the probability density $f$ estimated for the samples populating the polytopes using Equation 5, denoted as $\hat{f}$. The conditions for choosing the Gaussian kernel parameters are:*

*1. The center of the kernel can be any point $z_r$ within the polytope $Q_r$ as $n \to \infty$,*

*2. The kernel bandwidth along any dimension $\sigma_r$ is any positive number always bounded by the polytope bandwidth $h_n$ as $n \to \infty$, i.e., $\sigma_r = C_r h_n$, where $0 < C_r \leq 1$.*

*Consider the following assumptions as well:*

*1. The polytope bandwidth $h_n \to 0$ as $n \to \infty$.*

*2. $n$ grows faster than the shrinkage of $h_n$, i.e., $nh_n \to \infty$ as $h_n \to 0$ in probability.*

*Given these assumptions, we have that as $n \to \infty$:*

$$\sup_{\mathbf{x} \in \mathbb{R}^d} |f(\mathbf{x}) - \hat{f}(\mathbf{x})| \to 0,$$

*where $|\cdot|$ denotes absolute value of the scalar it operates on.*

*Proof.* Please see Appendix A for the proof. $\qquad\square$

**Theorem 2** (Asymptotic OOD Convergence). *Given $n$ independent and identically distributed training samples $\{(\mathbf{x}_i, y_i)\}_{i=1}^n$, we define the distance of an inference point $\mathbf{x}$ from the training points as:*

$$d_\mathbf{x} = \min_{i=1,\cdots,n} \|\mathbf{x} - \mathbf{x}_i\|. \tag{9}$$

*Here $\|\cdot\|$ denotes a suitable distance measure as mentioned in Equation 4. Given non-zero and bounded bandwidth of the Gaussians, then we have almost sure convergence for $\hat{g}_y$ as: $\hat{g}_y(\mathbf{x}) \xrightarrow{as} \hat{P}_Y(y)$ as $d_\mathbf{x} \to \infty$.*

*Proof.* Please see Appendix A for the proof. □

**Corollary 2.1.** *Given the conditions in Theorem 1 and 2, we have:*

$$\max_{y \in \mathcal{Y}} \sup_{\mathbf{x} \in \mathbb{R}^d} |g_y(\mathbf{x}) - \hat{g}_y(\mathbf{x})| \to 0.$$

*Proof.* Using the law of large numbers, we have $\hat{P}_Y(y) = \frac{n_y}{n} \xrightarrow{as} P_Y(y)$ as $n_y \to \infty$. The rest of the proof follows from Theorem 1 and 2. □

# 5 Model Parameter Estimation

## 5.1 Gaussian Kernel Parameter Estimation

Theorem 1 implies that the Gaussian kernel parameters need to maintain two key properties. We use the training data within the polytopes to estimate the Gaussian parameters in a way that we asymptotically satisfy the above two conditions for consistency. To satisfy the first condition, we set the kernel center as:

$$\hat{\mu}_r = \frac{1}{n_r} \sum_{i=1}^n \mathbf{x}_i \mathbb{1}(\mathbf{x}_i \in Q_r). \tag{10}$$

Note that $\hat{\mu}_r$ in Equation 10 resides always within the corresponding polytope $Q_r$. For improving the estimates for the kernel bandwidth, we incorporate the samples from other polytopes $Q_s$ based on the similarity $w_{rs}$ between $Q_r$ and $Q_s$. Moreover, We constrain our estimated Gaussian kernels to have diagonal covariance matrix. We use weighted likelihood estimation to estimate the variance $\Sigma_r$ for a particular polytope $Q_r$. For simplicity, we will describe the estimation procedure for $w_{rs}$ later. The weighted likelihood estimation for $\Sigma_r$ can be written as:

$$\hat{\Sigma}_r = \underset{\Sigma}{\arg\min} - \sum_{i=1}^n \sum_{s \in \mathcal{P}} w_{rs} \mathbb{1}(\mathbf{x}_i \in Q_s) \log \mathcal{G}(\mathbf{x}_i; \hat{\mu}_r, \Sigma) + \lambda \|\Sigma^{-1}\|_F^2, \tag{11}$$

where we regularize the Frobenius norm of precision matrix $\Sigma^{-1}$ so that $\Sigma$ does not become singular and $\lambda$ is the regularization parameter. By solving Equation 11, we find:

$$\hat{\Sigma}_r = \frac{\sum_{s \in \mathcal{P}} \sum_{i=1}^n w_{rs} \mathbb{1}(\mathbf{x}_i \in Q_s)(\mathbf{x}_i - \hat{\mu}_r)(\mathbf{x}_i - \hat{\mu}_r)^\top + \lambda I_d}{\sum_{s \in \mathcal{P}} \sum_{i=1}^n w_{rs} \mathbb{1}(\mathbf{x}_i \in Q_s)}, \tag{12}$$

where, $I_d$ is a $d$ dimensional identity matrix. However, we want $\Sigma_r$ to be estimated based on the samples within $Q_r$ so that the second condition for the Gaussian parameters is satisfied. Therefore, as $n \to \infty$ and $h_n \to 0$, the estimated weights $w_{rs}$ should satisfy the condition:

$$w_{rs} \to \begin{cases} 0, & \text{if } Q_r \neq Q_s \\ 1, & \text{if } Q_r = Q_s. \end{cases} \tag{13}$$

We need Condition 13 as we will be only using the data within the polytope $Q_r$ as $n \to \infty$ to estimate the Gaussian bandwidth and the estimated Gaussian bandwidth will be bounded by the polytope bandwidth. Additionally, we use weighted samples to replace the ratio $\frac{n_{ry}}{n_y}$ in Equation 5 as:

$$\frac{\tilde{w}_{ry}}{\tilde{w}_y} = \frac{\tilde{w}_{ry}}{\sum_{r \in \mathcal{P}} \tilde{w}_{ry}} = \frac{\sum_{s \in \mathcal{P}} \sum_{i=1}^n w_{rs} \mathbb{1}(\mathbf{x}_i \in Q_s) \mathbb{1}(y_i = y)}{\sum_{r \in \mathcal{P}} \sum_{s \in \mathcal{P}} \sum_{i=1}^n w_{rs} \mathbb{1}(\mathbf{x}_i \in Q_s) \mathbb{1}(y_i = y)}. \tag{14}$$

Note that if we satisfy Condition 13, then we have $\frac{\tilde{w}_{ry}}{\tilde{w}_y} \to \frac{n_{ry}}{n_y}$ as $n \to \infty$. Therefore, we modify Equation 5 as:

$$\tilde{f}_y(\mathbf{x}) = \frac{1}{\tilde{w}_y} \sum_{r \in \mathcal{P}} \tilde{w}_{ry} \mathcal{G}(\mathbf{x}; \hat{\mu}_r, \hat{\Sigma}_r) \mathbb{1}(r = \hat{r}_{\mathbf{x}}^*), \tag{15}$$

where $\hat{r}_{\mathbf{x}}^* = \mathrm{argmin}_r \|\hat{\mu}_r - \mathbf{x}\|$. Below, we describe how we estimate $w_{rs}$ for KGF and KGN .

## 5.2   Kernel Generative Forest

Consider $T$ number of decision trees in a random forest trained on $n$ i.i.d training samples $\{(\mathbf{x}_i, y_i)\}_{i=1}^n$. Each tree $t$ partitions the feature space into $p_t$ polytopes resulting in a set of polytopes: $\{\{Q_{t,r}\}_{r=1}^{p_t}\}_{t=1}^T$. The intersection of these polytopes gives a new set of polytopes $\{Q_r\}_{r=1}^p$ for the forest. For any point $\mathbf{x}_r \in Q_r$, we push every other sample point $\mathbf{x}_s \in Q_s$ down the trees. Now, we define the weight $w'_{rs}$ as:

$$w'_{rs} = \frac{t_{rs}}{T}, \tag{16}$$

where $t_{rs}$ is the total number trees $\mathbf{x}_r$ and $\mathbf{x}_s$ end up in the same leaf node. Note that $0 \le w'_{rs} \le 1$. *If the two samples end up in the same leaves in all the trees, they belong to the same polytope, i.e. $Q_r = Q_s$.*

In short, $w'_{rs}$ is the fraction of total trees where the two samples follow the same path from the root to a leaf node. We exponentiate $w'_{rs}$ with a suitable function of $n$ which grows with $n$ so that Condition 13 is satisfied:

$$w_{rs} = (w'_{rs})^{O(n)}. \tag{17}$$

## 5.3   Kernel Generative Network

Consider a fully connected ReLU-net trained on $n$ i.i.d training samples $\{(\mathbf{x}_i, y_i)\}_{i=1}^n$. We have the set of all nodes denoted by $\mathcal{N}_l$ at a particular layer $l$. We can randomly pick a node $n_l \in \mathcal{N}_l$ from $\mathcal{A}_l$ at each layer $l$, and construct a sequence of nodes starting at the input layer and ending at the output layer which we call an **activation path**: $m = \{n_l \in \mathcal{N}_l\}_{l=1}^L$. Note that there are $N = \Pi_{i=1}^L |\mathcal{N}_l|$ possible activation paths for a sample in the ReLU-net, where $|\cdot|$ denotes the cardinality or the number of elements in the set. We index each path by a unique identifier number $z \in \mathbb{N}$ and construct a sequence of activation paths as: $\mathcal{M} = \{m_z\}_{z=1,\cdots,N}$. Therefore, $\mathcal{M}$ contains all possible activation pathways from the input to the output of the network.

While pushing a training sample $\mathbf{x}_i$ through the network, we define the activation from a ReLU unit at any node as '1' when it has non-negative input and '0' otherwise. Therefore, the activation indicates on which side of the affine function at each node the sample falls. The activation for all nodes in an activation path $m_z$ for a particular sample creates an **activation mode** $a_z \in \{0, 1\}^L$. If we evaluate the activation mode for all activation paths in $\mathcal{M}$ while pushing a sample through the network, we get a sequence of activation modes: $\mathcal{A}_r = \{a_z^r\}_{z=1}^N$. Here $r$ is the index of the polytope where the sample falls in.

*If the two sequences of activation modes for two different training samples are identical, they belong to the same polytope.* In other words, if $\mathcal{A}_r = \mathcal{A}_s$, then $Q_r = Q_s$. This statement holds because the above samples will lie on the same side of the affine function at each node in different layers of the network. Now, we define the weight $w'_{rs}$ as:

$$w'_{rs} = \frac{\sum_{z=1}^N \mathbb{1}(a_z^r = a_z^s)}{N}. \tag{18}$$

Note that $0 \le w'_{rs} \le 1$. In short, $w'_{rs}$ is the fraction of total activation paths which are identically activated for two samples in two different polytopes $r$ and $s$. We exponentiate the weights using Equation 17.

Pseudocodes outlining the two algorithms are provided in Appendix C.

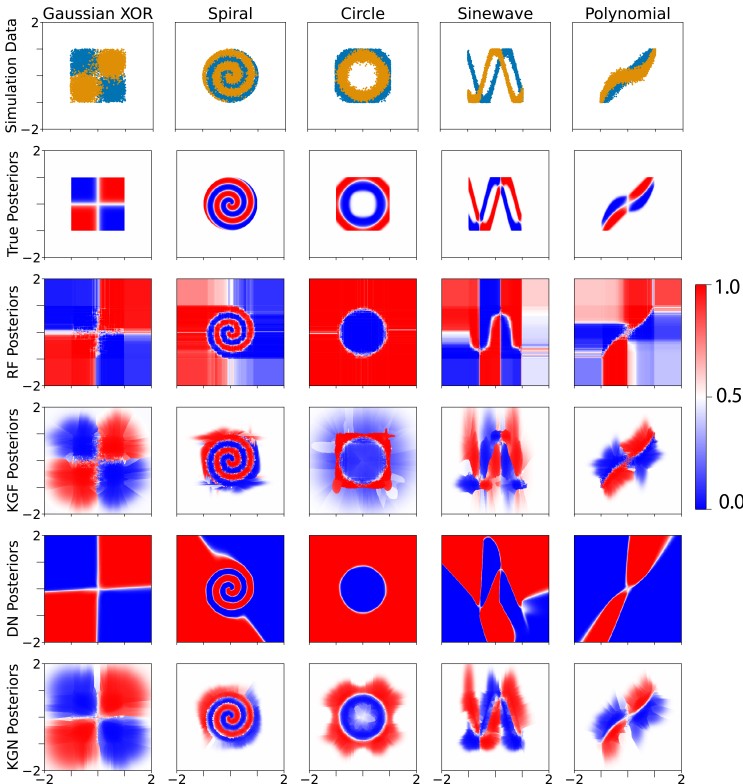

Figure 1: **Visualization of true and estimated posteriors for class 0 from five binary class simulation experiments.** *Row 1*: 10,000 training points with 5,000 samples per class sampled from 5 different simulation setups for binary class classification. The class labels are indicated by yellow and blue colors. *Row 2*: True class conditional posteriors. *Row 3*: Estimated posteriors from random forest. *Row 4*: Estimated posteriors from KGF. *Row 5*: Estimated posteriors from Deep-net. *Row 6*: Estimated posteriors from KGN. The posteriors estimated from KGN and KGF are better calibrated for both in- and out-of-distribution regions compared to those of their parent algorithms.

## 6  Experimental Results

We conduct several experiments on two dimensional simulated datasets and OpenML-CC18 data suite [21] [1] to gain insights on the finite sample performance of KGF and KGN. The details of the simulation datasets and hyperparameters used for all the experiments are provided in Appendix B. For the simulation setups, we use classification error, hellinger distance [22, 23] from the true class conditional posteriors and mean max confidence or posterior [4] as performance statistics. While measuring in-distribution calibration for the datasets in OpenML-CC18 data suite, as we do not know the true distribution, we used adaptive calibration error as defined by Nixon et al. [24] with a fixed bin number of $R = 15$ across all the datasets. Given $n$ OOD samples, we define OOD calibration error to measure OOD performance for the benchmark datasets as:

$$\left| \frac{1}{n} \sum_{i=1}^{n} \max_{y \in \mathcal{Y}} (\hat{P}_{Y|X}(y|\mathbf{x}_i)) - \max_{y \in \mathcal{Y}} (\hat{P}_Y(y)) \right|.$$

### 6.1  Simulation Study

Figure 1 top row shows 10000 training samples with 5000 samples per class sampled within the region $[-1, 1] \times [-1, 1]$ from the five simulation setups described in Appendix B. Therefore, the empty annular region between $[-1, 1] \times [-1, 1]$ and $[-2, 2] \times [-2, 2]$ is the low density or OOD

---

[1] https://www.openml.org/s/99

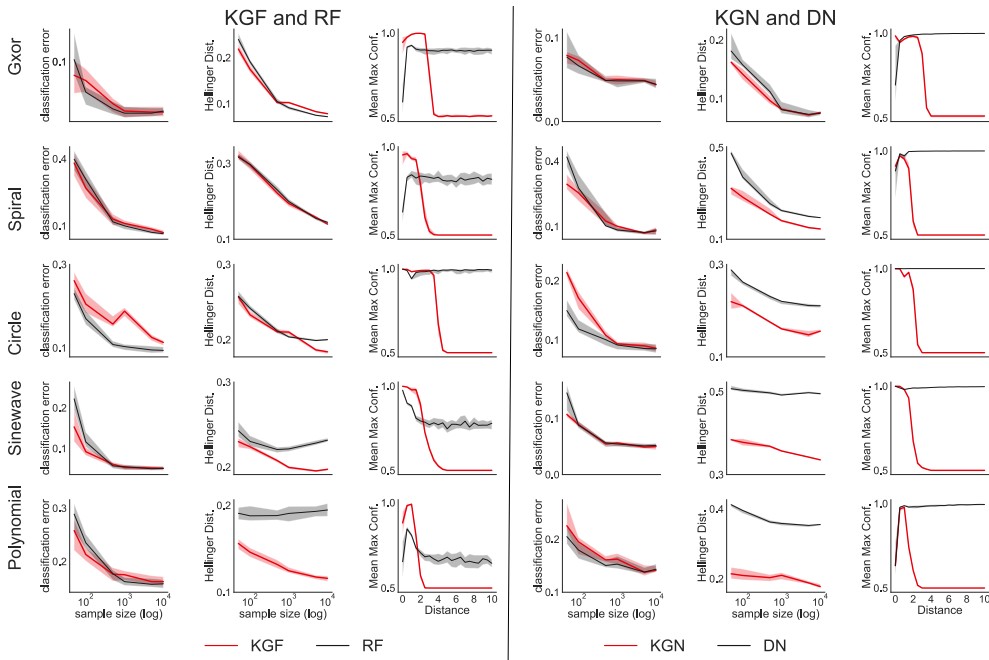

Figure 2: **Classification error, Hellinger distance from true posteriors, mean max confidence or posterior for the simulation experiments.** The median performance is shown as a dark curve with shaded region as error bars showing the 25-th and the 75-th percentile. KGF (**Left block**) and KGN (**Right block**) improve both in- and out-of-distribution calibration of their respective parent algorithms while maintaining nearly similar classification accuracy on the simulation datasets.

region in Figure 1. The corresponding true posteriors $\mathbb{P}[Y = 0|X = \mathbf{x}]$ are shown in the second row of Figure 1. As shown in Row 3 and 5, RF and DN are really good at estimating the high density regions of training distribution. However, they overestimate the posteriors in the low density regions of training distribution. Row 4 and 6 of Figure 1 show KGF and KGN improves the posterior estimation specially in the low density of the training distribution or OOD regions of the feature space. Because of axis aligned split in random forest RF and thereby, KGF are less efficient in learning non-linear decision boundaries like spiral, circle and sinewave simulations than ReLU-net and KGN. Figure 2 quantifies the performance of the algorithms which are visually represented in Figure 1. KGF and KGN maintain similar classification accuracy to those of their parent algorithms. We measure hellinger distance from the true distribution for increasing training sample size within $[-1, 1] \times [-1, 1]$ region as an index for in-distribution calibration. Column 2 of left and right block in Figure 2 show KGF and KGN are better at estimating the high density region of training distribution compared to their parent methods. For measuring OOD performance, we normalize the training data by the maximum of their $l2$ norm so that the training data is confined within a unit circle. For inference, we sample 1000 inference points uniformly from a circle where the circles have increasing radius and plot mean max posterior for increasing distance from the origin. Therefore, for distance up to 1 we have in-distribution samples and distances farther than 1 can be considered as OOD region. As shown in Column 3 of Figure 2, mean max posteriors or confidence for KGF and KGN converge to the maximum of the class priors, i.e., $0.5$ as we go farther away from the training data origin.

## 6.2 Benchmark Data Study

We used OpenML-CC18 data suite for benchmark dataset study. We exclude any dataset which contain categorical features or NaN values and conduct our experiments on 46 datasets with varying dimensions and sample sizes. For the OOD experiments, we follow a similar setup as that of the simulation data. We normalize the training data by their maximum $l2$ norm and sample 1000 testing samples uniformly from each hypersphere where the hyperspheres have increasing radius starting from 1 to 5. Figure 3 shows the summary of performance of the algorithms. The extended results for

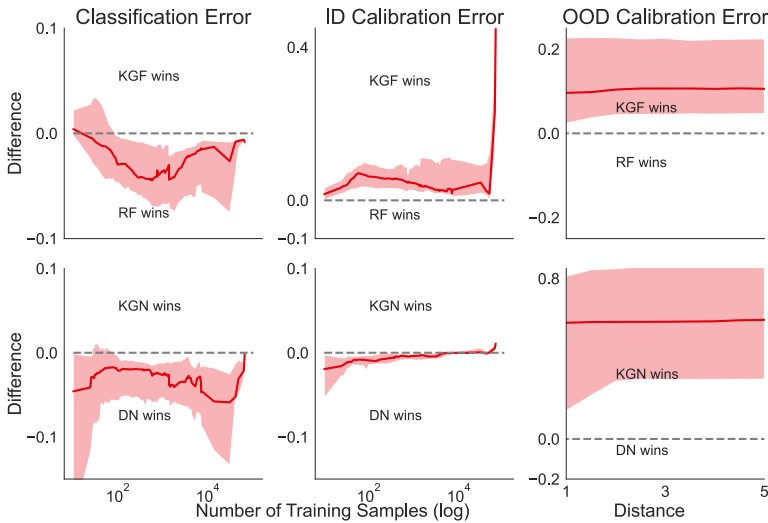

Figure 3: **Performance summary of** `KGF` **and** `KGN` **on OpenML-CC18 data suite.** The dark red curve in the middle shows the median of performance on 46 datasets. The shaded region shows the error bar consisting of the 25-th and the 75-th percentile of the performance statistics. **Left:** `KGF` and `KGN` maintains performance close to their parent algorithms for classification. **Middle:** `KGF` significantly improves the in-distribution calibration for random forest and `KGN` improves RELU-net's in-distribution calibration for high training sample sizes. **Right:** Both of the proposed approaches yield highly calibrated confidence in the OOD region.

each dataset is shown separately in appendix Figure 4, 5, 6, 7, 8 and 9. Figure 3 left column shows on average `KGF` and `KGN` has nearly similar classification accuracy to their respective parent algorithm. However, according to Figure 3 middle column, `KGF` improves the in-distribution calibration for random forest by a huge margin. On the contrary, `KGN` maintains similar in-distribution calibration performance to that of its parent RELU-net. Most interestingly, Figure 3 right column `KGN` and `KGF` improves OOD calibration of their respective parent algorithms by a huge margin.

## 7    Discussion

In this paper, we convert deep discriminative models to deep generative models by replacing the affine function over the polytopes in the discriminative models with a Gaussian kernel. This replacement of affine function results in better in- and out-of-distribution calibration for our proposed approaches while maintaining classification accuracy close to the parent algorithm. Theoretically, we show under certain conditions our approaches asymptotically converge to the true training distribution and this establishes confidence calibration for learning algorithms in in- and out-of-distribution regions as a continuum rather than two different problems.

For a feature space densely partitioned with small polytopes, we can use Euclidean distance metric in Equation 4. This is because the Euclidean manifold approximation holds locally for the corresponding polytope with index $r_{\mathsf{x}}^*$. On the contrary, for a feature space partitioned with large polytopes, Euclidean distance measure may be a wrong notion of distance, specially when the underlying manifold is non-Euclidean. Note that the indicator function in Equation 5 and pruning of the unpopulated polytopes result in an enlargement of the polytopes in our proposed method compared to that of the parent model in Equation 3. Therefore, our proposed approach while using Euclidean metric in Equation 4 may have less classification accuracy compared to that of its parent algorithm. A correct measure of distance in all the cases including the aforementioned non-Euclidean one would be the geodesic distance as explored by Madhyastha et al. [25]. We will explore convolutional neural nets (CNN) trained on image and language benchmark datasets using geodesic distance in Equation 4 in our future work. Additionally, the proposed approach needs benchmarking against other calibration approaches in the literature which are mainly based on image datasets and we will pursue the benchmarking task in our future work.

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
