# OpenReview forum: "Deep Discriminative to Kernel Generative Networks for Calibrated Inference"
_NeurIPS.cc/2023/Conference — Submitted to NeurIPS 2023_

### Official Review · Reviewer_Up6z · 2023-06-20

**Soundness:** 3 good
**Presentation:** 3 good
**Contribution:** 3 good
**Rating:** 4
**Confidence:** 3

**Summary:**

The paper presents an algorithm to learn an auxiliary model. The goal is to enhance the out-of-distribution calibration performance while maintaining the in-distribution performance of a base discriminative model. This base model is either based on a random forest or a multi-layer perceptron. The algorithm follows three main step: Firstly, it constructs an adjacency matrix using the training samples. The entries of this matrix depend on the "rule/activation patterns" of the corresponding discriminative model. Secondly, it performs clustering to divide the input space into regions. Lastly, for each region, the algorithm uses a Gaussian kernel to regress the output of the original model (by learning the center and the covariance matrix). The algorithm reaches convergence to the ground truth posterior distribution in the limit of infinite training data. The experiments are conducted on two-dimensional toy data and datasets from the OpenML-CC18 Benchmark Suites, thus demonstrating the algorithm's superior performance in terms of out-of-distribution calibration compared to the base models. Furthermore, it achieves comparable accuracy and in-distribution calibration performance to the base models.

**Strengths:**


1. The algorithm to identify the partition induced by a classifier (random forest, MLP) on the input space is original and novel. Indeed, it can help to shed light on the calibration properties of existing classifiers (**Originality**)
2. Asymptotic convergence of the algorithm to the true posterior is guaranteed in the infinite sample regime (**Soundness**).
3. Overall, the paper is clear (**Clarity**). The text could be improved by providing more details about the algorithm, rather than focusing on the discussion between discriminative and generative models, which seems a bit out of scope.
4. Code is available, but I haven’t tried to reproduce the experiments (**Reproducibility**)


**Weaknesses:**


1. It’s unclear how the algorithm performs in the finite sample regime and in high dimensions and how these two quantities are related. The scope of the analysis is therefore limited to a regime with infinite data (**Significance**). In absence of an analysis on the number of dimensions, the authors could provide an experimental analysis with higher dimensional datasets, like CIFAR10 or CIFAR100.
2. Previous works have shown that neural networks are over-confident/wrong on OOD data [1] and have suggested a set of experiments to check the phenomenon. Providing an analysis on such experimental settings might strengthen the work (**Quality**) - See Questions.
3. The authors could relate the partitioning technique with the literature on approximation based on splines (**Quality**) - See Questions.



**Questions:**


**Major questions:**
1. [1] provides an interesting insight on OOD classification. Specifically, when training on object datasets like CIFAR-10 and CIFAR-100, neural models assign high likelihood also to SVHN data. What does it happen when using the proposed algorithm? Does it overcome the weaknesses of neural approaches or does it preserve the same properties?
2. Considering that deep networks partition the input space with a number of regions that grow exponentially with network depth [2,3], how do you choose the number of clusters in the algorithm, to ensure high regression fidelity (in terms of accuracy and in-distribution calibration) with the base model?
3. How reliable is the analysis in the finite sample regime and in high dimensions?
4. It is not clear why the proposed approach is a generative model (Eq. (7) is never used. Indeed, the algorithm only relied on Eq. (15)). Can you please elaborate on that?
5. In Figure 2 (row 3, column 1) and in Figure 3 (column 1), the base model achieves higher accuracy with a larger number of samples. This seems counterintuitive. Do you have an explanation for this phenomenon?
6. What is the impact of pruning on the performance?

**Minor questions:**
1. In Eq. (6), what is the purpose of the bias and why should the bias vanish for infinite number of samples?
2. In Eq. (18), the computation of weights is based on the number of equivalent paths. Is there an intuitive explanation for preferring this solution over the one which simply counts the number of activated/disabled neurons in the network?

**References**
- [1] Do Deep Generative Models Know What They Don’t Know? ICLR 2019
- [2] On the Number of Linear Regions of Deep Neural Networks. NeurIPS 2014
- [3] A Spline Theory of Deep Networks. ICML 2018

**Limitations:**

The work is limited in the analysis (input dimensionality, class of neural network models).

---

> ### Author Rebuttal · Authors · 2023-08-10
>
> - **[1] provides an interesting insight on OOD classification. Specifically, when training on object datasets like CIFAR-10 and CIFAR-100, neural models assign high likelihood also to SVHN data. What does it happen when using the proposed algorithm? Does it overcome the weaknesses of neural approaches or does it preserve the same properties?**
>
> We have conducted additional vision experiments following this suggestion by the reviewer.
>
> - **Considering that deep networks partition the input space with a number of regions that grow exponentially with network depth [2,3], how do you choose the number of clusters in the algorithm, to ensure high regression fidelity (in terms of accuracy and in-distribution calibration) with the base model?**
>
> Please note Equation 5 and 15, Line 114 where we mention that we only consider the polytopes which are populated by the training data. Therefore, the number of polytopes considered in our approach is upper bounded by the number of training data. The proposed model loses regression fidelity in terms of accuracy for sparsely populated polytopes in high dimensional space. However, as shown in the additional experiment in the attached PDF, the loss in accuracy can be avoided using Geodesic distance in Equation 4.
>
> - **How reliable is the analysis in the finite sample regime and in high dimensions?**
>
> We have addressed this concern in the global response.
>
> - **It is not clear why the proposed approach is a generative model (Eq. (7) is never used. Indeed, the algorithm only relied on Eq. (15)). Can you please elaborate on that?**
>
> We apologize for emphasizing more on the generative aspect in the abstract, which may have deviated the reviewers from the main point of the paper which is `calibration’. We will revise the abstract in the final version without mentioning the generative aspect of the proposed approach. However, Equation 15 is an approximation of Equation 5. After describing Equation  15, we need to use the approximation in Equation 6, 7 and 8, respectively. We have fixed this in our revised draft. To sample from the estimated density, one can pick a kernel with probability $\frac{n_{ry}}{n_r}$ and sample from the corresponding Gaussian distribution. We discuss the possibility of sampling from the proposed model in the discussion.
>
> - **In Figure 2 (row 3, column 1) and in Figure 3 (column 1), the base model achieves higher accuracy with a larger number of samples. This seems counterintuitive. Do you have an explanation for this phenomenon?**
>
> The loss in accuracy in Figure 2 (row 3, column 1) can be explained using Figure 1 (row 4, column 3). The posteriors learned by KGF are noisy because of the axis aligned split used in the parent random forest at each node. This phenomenon has been mitigated by KGN (Figure 1 row 6, column 3) as the parent deep-net can implement non-linear decision boundaries. The loss of accuracy in Figure 3 (column 1) is due to the presence of high dimensional datasets in OpenML which was explained in the Discussion section. However, we have pursued additional experiments using Geodesic distance which can further elaborate about the loss of accuracy in Figure 3 (column 1).
>
> - **What is the impact of pruning on the performance?**
>
> The impact of pruning has been discussed in the Discussion section and a potential solution using Geodesic distance in Equation 4 was proposed which we have pursued in the additional experiments attached in the global response.
>
> - **In Eq. (6), what is the purpose of the bias and why should the bias vanish for infinite number of samples?**
>
> Theorem 1 is derived using the estimated density in Equation 5 (mentioned in the statement) which has no bias term. For Theorem 1 to be true for Equation 6, the bias term in Equation 6 should vanish for an infinite number of samples. As shown in Appendix A.2 (proof of Theorem 2), the bias term is necessary for Theorem 2 to be true. We will clarify this further in the final draft.
>
> - **In Eq. (18), the computation of weights is based on the number of equivalent paths. Is there an intuitive explanation for preferring this solution over the one which simply counts the number of activated/disabled neurons in the network?**
>
> In Section 7.3 of [1], the authors proposed a similarity measure from each layer of the network based on counting the number of activated/disabled neurons in the network. However, it is unclear how to combine the similarity from different layers into a single score while considering the order of the layers. This is because the lower layers encode base features and the higher layers gradually encode more complex features. One should consider this hierarchy of information from layers of the network while measuring the similarity score. However, our proposed approach inherently considers the order of activations from different layers resulting in less loss of information. The effectiveness of the proposed  similarity measure is further illustrated in the new additional high dimensional experiments where the model preserves the accuracy of the parent model when Geodesic distance is used as the similarity measure. Moreover, as illustrated in Algorithm 2 and 3 of the appendix, the similarity measure calculation in deep-nets is roughly analogous to that of random forests.
>
> [1] A Spline Theory of Deep Networks. ICML 2018.

---

> > ### Comment · Reviewer_Up6z · 2023-08-15
> > **Thank you for the Answer**
> >
> > Thank you for the clarifications and the additional experiments. I went through the whole rebuttal and I share the same feeling of other reviewers, namely that major effort is required by the authors to make the paper complete especially in terms of experiments.
> >
> > Specifically, the additional experiments provided by the authors with multi-variate Gaussians highlight the fact that the original proposed solution did not deal well with the curse of dimensionality. The authors subsequently proposed a modification to it by replacing the Euclidean distance with a geodesic one inspired by the weights computed in the kernel generative framework. While I appreciate this new solution and that this suggests to scale better to higher dimension, a careful analysis identifying its limitations is currently missing. Additionally, the experiments provided in Figure 5 of the attached pdf focus on an unconventional setting (from the continual learning literature, where the discriminative model is trained on binary classification tasks). I suggest the authors to properly run the evaluation in the multi-class setting following the same methodology used in [1] (as highlighted also by reviewer fq3V), [2] and [3]. This would indeed resolve the issue about missing baseline and also properly address my weakness 2.
> >
> > Overall, I think that the proposed solution is interesting and has the potential to make a good contribution to the literature of ID, OOD calibration. However, in its current form the work is not complete. For this reason, I update my score to 4.
> >
> > [1] Why ReLU networks yield high-confidence predictions far away from
> > the training data and how to mitigate the problem CVPR 2019
> >
> > [2] Do Deep Generative Models Know What They Don’t Know? ICLR 2019
> >
> > [3] Your Classifier is Secretly an Energy Based Model and You Should Treat it Like One. ICLR 2020

---

### Official Review · Reviewer_fq3V · 2023-07-04

**Soundness:** 3 good
**Presentation:** 3 good
**Contribution:** 3 good
**Rating:** 4
**Confidence:** 4

**Summary:**

This paper tackles the ID and OOD problem by proposing Kernel Generative Forest and Kernel Generative Network for estimating the similarity $w_{rs}$ and in turn approximate the class-conditional density.

The main contributions include: 1) theoretical results of the convergence of the approximated class-conditional density under certain conditions. 2) the similarity measure with KGF and KGN between the polytopes.

**Strengths:**

This paper proposed a novel idea of estimating ID and OOD by approximating the class-conditional density. For that, the similarities between polytopes of the input space when data falls into is estimated. Both theoretical and empirical results show that the proposed model can successfully tell when test data is OOD while preserving the classification accuracy.

The paper is overall well written.

This paper has the potential of contributing to the community.

**Weaknesses:**

The proposed method is highly related to [4], but in the experimental part there is no comparison to any of the recent methods such as [4] (except for the "parent algorithms" RF and DN). At least MMC is also reported in [4]. The readers will also benefit if a detailed discussion of similarity and difference between [4] is provided.

The background and the proposed method is well illustrated, but the demonstration of the experimental results is less clear.  e.g. 1) In fig 1 the simulated yellow points overwrite the blue points and therefore do not match the true posteriors. 2) The "Difference" in fig 3 is hard to understand, as well as the legends of "wins". What is the reason not plotting like Fig 2?



**Questions:**

 - What is $A_l$ in line 200 on page 6?

 - in Theorem 1, it requires the hypercubes to be of the same size. I am not sure if my understanding is correct or not, how can one partition e.g. the $R^1$ into 3 hypercubes $(-\infty, a), [a,b), [b, \infty)$ of same size?

 - what is "median performance"of classification error on the simulated datasets? Isn't the classification error calculated from the whole test set at once?



**Limitations:**

The paper discussed the limitations of the adopted Euclidean distance measure.

---

> ### Author Rebuttal · Authors · 2023-08-10
>
> - **The proposed method is highly related to [4], but in the experimental part there is no comparison to any of the recent methods such as [4] (except for the "parent algorithms" RF and DN). At least MMC is also reported in [4]. The readers will also benefit if a detailed discussion of similarity and difference between [4] is provided.**
>
> We have addressed this concern in the global response.
>
> - **In fig 1 the simulated yellow points overwrite the blue points and therefore do not match the true posteriors.**
>
> We apologize for being unclear. We will clarify the simulation more in the revised draft. Note that the first row in Figure 1 represents the samples which are randomly sampled according to the class conditional posterior in the second row. Please note that the posteriors are 0.5 at the junction of two class boundaries in the second row, meaning that samples from both the classes are equally likely to be sampled in those regions. This is why the yellow points and the blue points are overlapping in the first row.
>
> - **The "Difference" in fig 3 is hard to understand, as well as the legends of "wins". What is the reason not plotting like Fig 2?**
>
> We are sorry Figure 3 is unclear, we tried to summarize results from 46 different experiments.  To be more clear, we will include a few examples in Figure 3 organized like Figure 2.  Note that Figure 2 does not have a single panel that summarizes across all simulations.  We chose to make one for the real data, rather than cherry-picking results, or relegating it to the appendix.  The summary statistic we chose was the difference of medians, because averaging across datasets did not make much sense to us.  We will explore other options also for the final draft, and include all 46 experiments, depicted as those in Figure 2, in the Appendix for greater clarity.
>
> - **What is $A_l$ in line 200 on page 6?**
>
> We apologize for this typo. We have taken $A_l$ off.
>
> - **In Theorem 1, it requires the hypercubes to be of the same size. I am not sure if my understanding is correct or not, how can one partition e.g. the R1 into 3 hypercubes (−∞,a),[a,b),[b,∞) of same size?**
>
> Please note that there is no upper bound in the total number of polytopes in the statement, i.e.,$\mathcal{P} =$ {$Q_1, Q_2, \cdots $} . We can divide $R^1$ into an infinite number of hypercubes of the same size. As shown in the derivation of Theorem 1 in the appendix, we only need to consider the polytopes which are populated by the training samples while deriving the theoretical results. We have included the ‘infinite number of hypercube’ term in the theorem statement to emphasize this point.
>
> - **what is "median performance"of classification error on the simulated datasets? Isn't the classification error calculated from the whole test set at once?**
>
> The simulation experiments were repeated 45 times for different training sample sizes and 1000 test samples. The median of the performance over 45 runs are reported in Figure 2. We will elaborate it in the caption for the final submission.

---

> > ### Comment · Reviewer_fq3V · 2023-08-12
> >
> > Many thanks for the clarification.
> >
> > My point on Figure 1 is that the authors plot the blue samples first and then the yellow samples overwrite the blue ones. Visually, the number of samples from each class are not equal.
> >
> > I still don't get the "wins" label in Figure 3. If I am not mistaken, the classification performance of the proposed model drops a lot compared to the base model.
> >
> > Theorem 1 needs improvement since it is confusing as it is now.
> >
> > Thanks for the additional comparison.
> > However, from Fig.5, it is not clear how they perform with the metrics of Classification Error, Hellinger Dist, and MMC w.r.t. sample size and distance, which are shown in Fig.2.
> > I am also a bit confused with the MMC on e.g. noise, why the proposed model outperforms most of the cases but fails when training on task 1? And for the baselines, I don't think the model would give higher MMC on noise than test sets. Could you provide some insights why this is happening?

---

> > > ### Author Response · Authors · 2023-08-14
> > > **Thanks for the thoughtful feedbacks!**
> > >
> > > We really appreciate the helpful feedbacks from reviewer fq3V. In what follows we try to address the concerns:
> > > - **My point on Figure 1 is that the authors plot the blue samples first and then the yellow samples overwrite the blue ones. Visually, the number of samples from each class are not equal.**
> > >
> > > We apologize that we did not understand the query completely. We will use lower alpha for the plots so that the samples are more transparent and sample size equality is more obvious.
> > >
> > > - **I still don't get the "wins" label in Figure 3. If I am not mistaken, the classification performance of the proposed model drops a lot compared to the base model.**
> > >
> > > We are sorry that Figure 3 is still unclear.  We understand the “wins” legend may be confusing. We will take the legend out and explain the plot further in the caption. We plotted (error RF - error KGF) along the Y-axis against different sample sizes along the X-axis. Any point above the dashed line along 0, indicates error RF is greater than error KGF. Hence KGF performs better if the plotted curve stays above the dashed line.
> > >
> > > - **Theorem 1 needs improvement since it is confusing as it is now.**
> > >
> > > We are sorry that Theorem 1 seems confusing. Theorem 1 considers a partition rule that partitions a continuous feature space $R^d$ into an infinite number of hypercubes each having the same size $h_n$. Assuming specific conditions are met, the Theorem demonstrates that the estimated density obtained through Equation 5 converges pointwise to the actual density over time. Note that we need a partition rule on $R^d$ that yields an infinite number of hypercubes so that they are of the same size. However, among the infinite number of hypercubes, Equation 5 uses only those hypercubes which are populated by the training data. We will add the above texts to the draft so that the theorem is more clear.
> > >
> > > - **However, from Fig.5, it is not clear how they perform with the metrics of Classification Error, Hellinger Dist, and MMC w.r.t. sample size and distance, which are shown in Fig.2.**
> > >
> > > We did not have enough time to run the vision experiment for different sample sizes as we did for Figure 3. Hence, we showed the vision results for the maximum training sample size available to us. We are currently running the above experiments and will add them in the final draft. However, we cannot calculate Helinger distance unless we know the true posterior distribution like we do in simulation datasets. We will calculate ECE instead.
> > >
> > > - **I am also a bit confused with the MMC on e.g. noise, why the proposed model outperforms most of the cases but fails when training on task 1?**
> > >
> > > Please note that Figure 5 presents two different approaches. One approach employs the Euclidean distance as described in Equation 4, while the other utilizes the geodesic distance. We agree with the reviewer's observation that the approach using Euclidean distance occasionally encounters issues when trained on Task 1. This decline in performance is notably alleviated when the proposed method incorporates the geodesic distance instead. From the vision experiment, it can be concluded that the Euclidean distance is not as effective as the geodesic distance in detecting the nearest Gaussian kernel in high dimensional feature space.
> > >
> > > - **And for the baselines, I don't think the model would give higher MMC on noise than test sets. Could you provide some insights why this is happening?**
> > >
> > > We really thank the reviewer for pointing this out. We checked the experiment code again and found that we sampled the noise samples from a Uniform distribution over $[0,255]^{w \times h \times c}$ (w=width, h=height, c=channel of the images). Unfortunately, we did not normalize the noise samples by 255. We apologize for testing on the noise samples without normalization. We reran the baseline algorithm with the corrected code. As the seeds were fixed for the experiments, nothing other than the noise row in the heatmap changed for the baseline algorithm. The MMC scores for the noise samples are: Task 1: 0.83, Task 2: 0.71, Task 3: 0.74, Task 4: 0.81, Task 5: 0.86. As predicted by the reviewer, the MMC for ACET on noise is lower than that of the test sets, but they are still higher than those of KGN-Geodesic. We will correct the plot in the revised draft.

---

> > > > ### Comment · Reviewer_fq3V · 2023-08-14
> > > >
> > > > Thanks for the answer and finding out the mistake in the additional experiment.
> > > >
> > > > I have changed my rating to 4

---

### Official Review · Reviewer_Quey · 2023-07-06

**Soundness:** 3 good
**Presentation:** 2 fair
**Contribution:** 2 fair
**Rating:** 4
**Confidence:** 3

**Summary:**

The paper proposes to improve OOD detection for deep discriminative models by replacing the affine function over the polytopes with a Gaussian kernel, leading to a method called kernel generative networks.
An estimation method is developed for the proposed method and some theoretical results on asymptotic convergence to the true distribution and to OOD.
Results based on simulations show that the proposed method can estimate distribution better than the parent algorithms and have benefits in terms of OOD detection or calibration in some cases.

**Strengths:**

### originality
The method seems to be novel.

### quality
The proposed method is intuitive and sound.
The work includes useful theoretical results for the properties of the proposed method.

### clarity
The abstract downplay the main focus of the paper or main benefit of the proposed method which is to improve OOD detection/calibration.
It makes me confused while reading it for the first time.

### significance
The proposed method is simple to understand and could potentially be a standard algorithm in popular ML libraries.

**Weaknesses:**

### quality
The results/experiments of the paper require more work.

While the paper says the proposed method "results in better in- and out-of-distribution calibration", the results in Figure 3 show a contradicting or mixed results in different cases.
As this is the main claim of the paper, it requires an in-depth discussion on why the results are mixed and when do we expect the proposed method outperforms the parent algorithm.

As improved OOD detection is part of the main claim, the paper should compare some existing unsupervised method like [1].

[1] Zhang, Mingtian, Andi Zhang, Tim Z. Xiao, Yitong Sun, and Steven McDonagh. "Out-of-distribution detection with class ratio estimation." arXiv preprint arXiv:2206.03955 (2022).

**Questions:**

What's the computational cost of the proposed method compared to the parent algorithm?

**Limitations:**

Some limitations in terms of the use of Euclidean distance are mentioned, followed by a proposal to fix it by using geodesic distance.
It feels to me that the paper should actually explore this considering the mixed results from the experiments section.

---

> ### Author Rebuttal · Authors · 2023-08-10
>
> - **While the paper says the proposed method "results in better in- and out-of-distribution calibration", the results in Figure 3 show a contradicting or mixed results in different cases.**
>
> We apologize for not being clear while describing the results in Figure 3. We will clarify the results in Figure 3 further in the final draft. Note that OpenMl comprises many high dimensional datasets. We have included additional experiments on a high dimensional simulation and vision datasets using geodesic distance which explains the loss of accuracy in Figure 3. However, as mentioned in Figure 1 of [1], shallow networks are well-calibrated for in-distribution. In our OpenML experiments, we used a relatively small network which can explain why KGN maintains similar in-distribution calibration to that of its parent algorithm for lower sample size. Conducting experiments with an overparameterized network will result in a large activation pattern which would slow down our algorithm with its current implementation. However, the number of nodes can be reduced significantly by pooling at each layer as proposed in [2]. To keep the content of the draft precise and concise, we will pursue overparameterized networks in a future work. We will add the above description in the Discussion section.
>
> - **As improved OOD detection is part of the main claim, the paper should compare some existing unsupervised method.**
>
> We have addressed this concern in the global response.
>
> - **What's the computational cost of the proposed method compared to the parent algorithm?**
>
> We have addressed this concern in the global response.
>
> [1] Guo, Chuan, et al. "On calibration of modern neural networks." International conference on machine learning. PMLR, 2017.
>
> [2] Olber, Bartłomiej, et al. "Detection of out-of-distribution samples using binary neuron activation patterns." Proceedings of the IEEE/CVF Conference on Computer Vision and Pattern Recognition. 2023.

---

> > ### Comment · Reviewer_Quey · 2023-08-21
> >
> > Thanks for the response and extra experiments.
> > However, my original concerns about mixed results still hold.
> > The authors have suggest to do some of the experiments as future work and the new experiments on high-dimensional data use some tricks that are not adequately discussed in the paper.
> > Although I believe the idea in the paper is interesting, I don't think it's ready for publication in its current form.
> > Therefore I keep my score as 4.

---

### Official Review · Reviewer_c3Ag · 2023-07-10

**Soundness:** 3 good
**Presentation:** 3 good
**Contribution:** 2 fair
**Rating:** 4
**Confidence:** 3

**Summary:**

The paper proposes a new method for confidence calibration in discriminative deep ReLU networks and random forests based on approximating the class-conditional density with Gaussian kernels.

**Strengths:**

- The proposed method is conceptually simple and fairly novel, and it does not require retraining the parent learner to work.
- Math and derivation are laid out clearly.
- The limitations of the work are adequately discussed.

**Weaknesses:**

- **Writing has room for improvement.** The writing can sometimes be unnecessarily long and convoluted. For example on lines 35-36: "However, one can adversarially manipulate an OOD sample where the model is less confident to find another OOD sample where the model is overconfident" and on lines 43-46: "The general idea for the generative group is to get likelihoods for a particular sample out of the generative models for both ID and OOD to do likelihood ratio test or control the likelihood for training distribution far away from the training data to detect OOD samples by thresholding." I would suggest the author break down these long sentences into shorter ones that are easier for the readers to parse.

- **Some potential problems are left unaddressed.** It is unclear to me how the proposed method would be able to overcome the curse of dimensionality, as the number of polytopes can scale exponentially with the number of neurons. I also find the claim that the proposed method converts discriminative networks into generative networks misleading. Although the paper does provide an expression for the class conditional density $\hat{f}_y(x)$, it seems highly nontrivial to sample from this unnormalized density. I wish the authors would clarify on this point.

- **No baselines in the experiments.** As the authors noted themselves, the experimental analyses in this paper are limited to comparisons between the original networks and the proposed kernel generative versions. Due to the lack of baselines, it is unknown whether the achieved improvements are significant or not, especially when they are obtained at the cost of classification accuracy as evident in figure 3.

Overall, I think that the methods proposed in the paper is potentially interesting, but more experiments needs to be done to demonstrate its practical effectiveness. I think the appeal of this work is really hindered by the lack of baseline comparisons and the limited analyses of the results, as well as some confusing/potentially misleading statements.

**Questions:**

- Can you clarify what is meant by "convert[ing] deep discriminative networks to kernel generative networks" in the abstract?
- How would the proposed method scale with dimensionality?
- What is the additional computational cost of estimating the parameters of the Gaussian kernels? How does this scale with model size/sample size?

**Limitations:**

The authors addressed the limitation that the Euclidean metric they used might not be the most suitable. Also, the authors are aware of the lack of comparison to other benchmark methods and commented that they will include it in future work.

---

> ### Author Rebuttal · Authors · 2023-08-10
>
> - **Writing has room for improvement.**
>
> We will thoroughly go over the text with an editor to ensure every sentence is clear and concise.
>
> - **It is unclear to me how the proposed method would be able to overcome the curse of dimensionality, as the number of polytopes can scale exponentially with the number of neurons.**
>
> Our methods will not overcome the curse of dimensionality, but it can partially mitigate it.  Because we only consider the polytopes populated by the training data, the number of polytopes is upper bounded by the training data sample size. This renders the dimensionality effectively always smaller than the training sample size.  We will modify the results section to clarify this point, using Figure 4 in the PDF to illustrate this point.
>
> - **Although the paper does provide an expression for the class conditional density f^y(x), it seems highly non trivial to sample from this unnormalized density.**
>
> We apologize for emphasizing a lot on the generative model in the abstract.  We will remove the generative aspect in the final version. That said, we are estimating normalized densities, so we could sample from them.  We will clarify this point in the discussion.
>
> - **No baselines in the experiments.**
>
> We have addressed this concern in the global response.
>
> - **Can you clarify what is meant by "convert[ing] deep discriminative networks to kernel generative networks" in the abstract?**
>
> We will remove this sentence from the abstract, as it is not demonstrated in the paper.  We will discuss the possibility of generating samples in the discussion.
>
> - **How would the proposed method scale with dimensionality?**
>
> We have addressed this concern in the global response.
>
> - **What is the additional computational cost of estimating the parameters of the Gaussian kernels? How does this scale with model size/sample size?**
>
> We have addressed this concern in the global response.

---

> > ### Comment · Reviewer_c3Ag · 2023-08-16
> >
> > I thank the authors for their response and I would like to say that I really appreciate their honesty. Unfortunately since the claim for "converting discriminative networks to generative networks" is largely vacuous and not a main result/focus of the paper, I stand by my judgement that this paper might have potential but in its current state is marginally below the bar for acceptance.

---

### Author Rebuttal · Authors · 2023-08-10

We extend our sincere gratitude to all the reviewers for their helpful suggestions and feedback, which we have incorporated to further improve our work. In particular, we pursued the suggestion from reviewer Quey to use geodesic distance and demonstrated its effectiveness with additional experiments. We have also run an experiment to demonstrate the effect of scaling input dimensionality on the kernel generative algorithms. These results are summarized in the attached PDF.

We address some shared concerns of the reviewers below.
-  **Abstract**: We apologize for emphasizing more on the generative aspect in the abstract, which may have deviated the reviewers from the main point of the paper which is `calibration’. We will revise the abstract in the final version without mentioning the generative aspect of the proposed approach.
- **Scaling of Accuracy with Dimensions**: As we have described in the discussion section, increasing the input dimensionality may have a detrimental effect on accuracy if we use Euclidean distance to find the nearest polytope. Inspired by the suggestion from reviewer Quey, we have used geodesic distance in a simulation experiment using the Trunk simulation data described in [1]. The binary class simulation is done using 2 multivariate Gaussians assigned to 2 different classes with their means being increasingly closer to each other in the higher dimensions. Thus, higher dimensions have increasingly less discriminative information. We have already discussed in the draft how to measure similarity between two samples $x_1 \in Q_r$ and $x_2 \in Q_s$ using $w_{rs}$. We have used $(1 - w_{rs})$ as the geodesic distance measure in Equation 4 (see [2] for a similar approach). The results from this experiment are shown in the attached PDF. The experiment demonstrates KGN-Geodesic and KGF-Geodesic scale similarly to that of their parent algorithms, overcoming the scaling problems of the Euclidean counterparts. Moreover, following the suggestion by reviewer Up6z, we have conducted an experiment on CIFAR-10 which shows similar results to that of the simulation experiment. For simplicity, we construct 5 different binary classification subtasks (e.g. Cats vs Dog classification) from CIFAR-10 akin to [3]. We collectively call these subtasks CIFAR-10 2X5. Five LeNet-5 models along with five KGN models were trained on each of these tasks. We have reported the mean max confidence (MMC) of the models in a heatmap along with their corresponding accuracies. Note that discriminating between the CIFAR-10 subtasks is much harder than distinguishing between CIFAR-10 and SVHN, as the images are semantically similar. In the vision experiments, it is evident that KGN-Geodesic not only maintains the accuracy of the parent model but also effectively distinguishes between ID and OOD samples. Conversely, other models fall significantly short in achieving this level of distinction.
- **No Baselines in the Experiments**: Following the suggestion by reviewer fq3V, we have compared our proposed method with ACET [4] which is an unsupervised SOTA OOD calibration method. The comparison is performed over the CIFAR-10 2X5 subtasks. Our experiment demonstrates that ACET fails to maintain OOD calibration over all the feature space even though it maintains nearly the same accuracy to the parent model. We will provide a detailed discussion about these experiments in our final draft.
- **Additional Computational Time**: The additional computational cost is dominated by the cost of calculating the adjacency matrix $W_{rs}$ which is $O(mn^2)$. Here m is the total number of nodes for ReLU-nets or total number of leaves for random forests. The number of nodes can be reduced significantly by pooling at each layer as proposed in [5] which we will pursue in future. However, we used all the nodes without pooling. One important point to note here is that the adjacency matrix computation can be easily parallelized which has been implemented in the kdcnn.py source code in the supplementary materials.

[1] Trunk, Gerard V. "A problem of dimensionality: A simple example." IEEE Transactions on pattern analysis and machine intelligence 3 (1979): 306-307.

[2] Madhyastha, Meghana, et al. "Geodesic forests." Proceedings of the 26th ACM SIGKDD International Conference on Knowledge Discovery & Data Mining. 2020.

[3] Zenke, Friedemann, Ben Poole, and Surya Ganguli. "Continual learning through synaptic intelligence." International conference on machine learning. PMLR, 2017.

[4] Matthias Hein, Maksym Andriushchenko, and Julian Bitterwolf. Why relu networks yield high-confidence predictions far away from the training data and how to mitigate the problem. In Proceedings of the IEEE/CVF Conference on Computer Vision and Pattern Recognition, pages 41–50, 2019.

[5] Olber, Bartłomiej, et al. "Detection of out-of-distribution samples using binary neuron activation patterns." Proceedings of the IEEE/CVF Conference on Computer Vision and Pattern Recognition. 2023.

---

> ### Author Response · Authors · 2023-08-14
> **Correction for wrong baseline results in the vision experiment**
>
> As kindly suggested by reviewer fq3V, we checked the vision experiment code again and found that we sampled the noise samples from a Uniform distribution over [0,255]^{w X h X c} (w=width, h=height, c=channel of the images). Unfortunately, we did not normalize the noise samples by 255. We apologize for testing on the noise samples without normalization. We reran the baseline algorithm with the corrected code. As the seeds were fixed for the experiments, nothing other than the noise row in the heatmap changed for the baseline algorithm. The MMC scores for the noise samples are: Task 1: 0.83, Task 2: 0.71, Task 3: 0.74, Task 4: 0.81, Task 5: 0.86. As predicted by the reviewer, the MMC for ACET on noise is lower than that of the other test sets, but they are still higher than those of KGN-Geodesic. We will correct the plot in the revised draft.

---

### Decision · Program_Chairs · 2023-09-21

**Decision:**

Reject

**Comment:**

All reviewers found a number of strengths in the paper, but after the discussion period there was still a consensus to reject in its current form. The proposed changes for an updated paper would constitute "major" corrections, and as AC I would suggest the revised paper goes through another round of review.